# Resilience at Work, Burnout, Secondary Trauma, and Compassion Satisfaction of Social Workers Amidst the COVID-19 Pandemic

**DOI:** 10.3390/ijerph19095500

**Published:** 2022-05-01

**Authors:** Anva Ratzon, Moshe Farhi, Navah Ratzon, Bruria Adini

**Affiliations:** 1Independent Researcher, Tel Aviv 6997801, Israel; 2Social Work Department, Tel-Hai College, Qiryat Shemona 1220800, Israel; moshefar@telhai.ac.il; 3Occupational Therapy Department, School of Health Professions, Sackler Faculty of Medicine, Tel Aviv University, Tel Aviv 6997801, Israel; navah@tauex.tau.ac.il; 4Department of Emergency & Disaster Management, School of Public Health, Sackler Faculty of Medicine, Tel Aviv University, Tel Aviv 6997801, Israel; adini@tauex.tau.ac.il

**Keywords:** work-resilience, social workers, COVID-19, burnout, compassion satisfaction, secondary trauma

## Abstract

Social workers during the COVID-19 pandemic are at risk due to exposure to varied populations in need, which may impact their resilience, burnout, secondary trauma, and compassion satisfaction. The study assessed resilience at work, burnout, secondary trauma, and compassion satisfaction among social workers in Israel during the first wave of the COVID-19 pandemic (May to June 2020). A convenience sample of 332 social workers (291 women (87.6%)) filled out an online, structured questionnaire that included demographics, a professional quality of life scale (ProQOL) (including three subscales), and resilience at work (RAW) (including seven subscales). The overall mean of the RAW was medium (M = 71, SD ± 8.9) compared to standardized scores. The mean scores of two of the subscales of the RAW, maintaining perspective and staying healthy, were low. The mean scores of the sub-scales of ProQOL were: compassion satisfaction was close to the 50th percentile (M = 48.25); burnout (M = 30.18) and secondary trauma (M = 26.27) were below the 25th percentile. Significant low to medium positive associations were found between all the dependent variables, except for staying healthy. A negative association was identified between compassion satisfaction and burnout, as well as between compassion satisfaction and secondary trauma. High levels of compassion satisfaction and contentment, low levels of secondary trauma, and having a managerial position were predicted to be 40% of the RAW. Lower levels of maintaining perspective, secondary trauma, and being younger predicted 27% of burnout. Higher levels of finding your calling, living authentically, maintaining perspective, interacting cooperatively, being older, and not being a manager predicted 58% of compassion satisfaction. Lower levels of burnout, maintaining perspective, and being younger predicted 36% of secondary trauma. As the COVID-19 pandemic still challenges most societies, policymakers should consider ways to integrate mechanisms that will enhance social workers’ resilience at work.

## 1. Introduction

Social work is a practice-based profession aimed at empowering individuals and communities through the enhancement of social change. Although implemented in communities worldwide, great diversity can be identified concerning the types of practices adopted, the varied organizations in which it operates, the target populations, the services provided, and the types of interventions that are commonly used (Arazi et al., 2020) [1]. Social workers are required to assist vulnerable populations routinely, but their importance is accentuated in the current COVID-19 crisis. The European Social Workers Union has stated that social workers are vital frontline responders in the campaign to contain COVID-19, as they facilitate the capacity of communities to protect themselves and others through physical distancing and social solidarity [2].

The social services workforce in general, and social workers in particular, provide crucial services during COVID-19 for diverse populations, including children, people with disabilities, the elderly, and more. During COVID-19, social workers provided services not only to individuals who were under their care prior to the pandemic but also to a large range of previously unknown populations who were substantially affected by its restrictions. These included the elderly, people with health risks, and wide groups that were negatively affected by the resulting economic crises [3,4,5]. Considering the social distancing regulations, social workers need to apply innovative methods and skills to provide the required services to clients and communities. Such supporting services encompass the provision of moral and instrumental support, mitigation, assistance to enable individuals to access services or advocate for their needs, therapy to enhance resilience and well-being, and more [6].

The work of social workers is characterized by numerous challenges and constraints in routine and, even more so, during adversities. This profession suffers from a lack of manpower, resources, and the budget required to manage many caseloads, which affect the ability of social workers to provide optimal care. Furthermore, in the past years, social workers have been exposed to negative media and social media coverage, which affect the social image of the profession, at times even leading to their being violently attacked [1,7,8]. Although there is an increasing need for social services during crises, throughout the COVID-19 pandemic, allocated budgets did not change, and manpower was not expanded, yet the work was even more demanding [9,10]. Social workers put themselves and their loved ones at risk of infection while they continue to work as frontline responders, meeting clients and organizations despite very limited access to personal protective equipment [11]. To meet this risk, some of the organizations temporarily ceased to provide services or, alternatively, used distant communication, but these solutions were found to impact the service effectiveness negatively [12]. Treatment methodologies were required to adapt to new concerns of the clients, such as anxiety concerning the virus, rising unemployment rates, heightened financial difficulties, and, most noted, loneliness. The social workers themselves face a joint changing of reality with their clients and the uncertainty of the stressful situation, which may complicate their capacity to provide a calm and reassuring atmosphere. In addition, social workers face ethical challenges, especially concerning equity in service provision, as well as the restricted ability to assist in situations of end-of-life, as they are unable to allow family members to be by the side of their dying loved ones [11,13]. Social workers have reported being unable to provide services as expected according to their ethical codes, agency policy, or regulations, most especially due to varied restrictions [6,14]. This may substantially impact their resilience at work and levels of burnout, partly resulting from secondary trauma.

Secondary trauma is the negative impact of indirect exposure to trauma on caregivers and therapists, such as social workers, who provide aid to patients after traumatic events. Caregivers and therapists are at risk of developing symptoms and reactions, such as suffering from post-traumatic stress disorder (PTSD) [15,16,17]. This risk is similar to other frontline workers, such as physicians and nurses, who were found to have a higher risk for mental health problems compared to the general population due to their exposure to COVID-19 patients [18]. Approximately 41% of healthcare workers were found to have symptoms of secondary trauma [19].

Ben-Porat [20] (2013) found a medium level of secondary trauma among social workers, with similar levels among those who work in domestic violence prevention centers and/or in welfare departments. Dagan et al. [17] (2016) found that over 50% of social workers working in the domain of child protection indicated that they have a high to severe level of secondary trauma. In another study that examined levels of secondary trauma among 412 social workers, low to moderate levels of secondary trauma were identified [21]. All the aforementioned studies used the secondary traumatic stress scale questionnaire [22], which includes 17 items, measuring intrusion, avoidance, and arousal symptoms associated with indirect exposure to traumatic events.

A study of a sample of 181 social workers conducted in the United States using the ProQOL scale during the COVID-19 pandemic found that approximately 50% of the respondents reported signs of secondary trauma at a medium level [23]. In a qualitative study conducted during the COVID-19 pandemic in California, in which 34 Master’s students of the Social Work Program were interviewed, most of the respondents expressed themes related to secondary trauma in their workplace [24]. Burnout is related to work conditions and a lack of coping resources. The symptoms of burnout can be presented as emotional (depression, compassion fatigue, and secondary trauma) or physical exhaustion (varied health conditions or illnesses). Burnout among social workers has been extensively studied, presenting high correlations between the social work profession and burnout syndromes, which are most frequently caused by it being a high-stress occupation [25,26,27]. Beyond the personal effects of burnout on social workers, it has also been shown that burnout negatively impacts professional functioning and leads to lower levels of quality of care for patients [28]. Furthermore, there are additional risk factors that contribute to the burnout of social workers derived from their low-income profession, which is characterized by a high level of workload, short deadlines, low appreciation, and limited resources, both for the social workers as service providers as well as for their organizations [29,30]. Conversely, enhancing psychological capital may serve as a protective measure for social workers against burnout and secondary trauma [31].

During the COVID-19 pandemic, the levels of stress and burnout increased for varied reasons, including work-related conditions [32]. Although many researchers have suggested that social workers are at a high risk of burnout [8,33], only a few studies have actually measured these levels during the COVID-19 pandemic. One such study, conducted in Spain using the Maslach Burnout Inventory [34], identified high levels of emotional exhaustion, depersonalization, and a reduced sense of personal accomplishment (all the constructs that demonstrate burnout) among social workers. Almost 70% of the social workers considered applying for psychotherapy due to COVID-19 [35]. In contrast, Holmes et al. [23] (2021) used the ProQOL scale and found that among 181 social workers screened, 63.71% expressed medium levels of burnout (M = 25.629, SD = 6.29, range 11–40). 

Although social workers are at risk of secondary trauma and burnout as a negative reaction to their work, they can also experience compassion satisfaction. Compassion satisfaction signifies positive feelings perceived by caregivers and service providers derived from helping people in need. Research has shown that compassion satisfaction is a protective occupation factor for varied professionals, such as social workers, therapists, and emergency responders. Compassion satisfaction can protect from stress, burnout, and secondary trauma [36,37,38]. In research conducted in Norway among child protection social workers, most of the workers were found to be moderately satisfied with their work [38]. Senreich et al. [39] (2020) assessed 6112 social workers from 13 different states in the United States and found that nearly 60% of them reported high levels of compassion satisfaction. They also found that the workplace environment’s wellness contributed to promoting compassion satisfaction. Similarly, during the COVID-19 pandemic, 99% of the social workers that were assessed reported a high level of compassion satisfaction [23]. Despite an extensive literature review, no additional studies on this subject were found.

As delineated, emotional and physical work experience and conditions can lead to burnout and secondary trauma but, in contrast, may also result in compassion satisfaction [37,40]. A negative correlation was found between secondary trauma and burnout compared to compassion satisfaction [38,40,41]. For example, Staudt and Williams [42] (2019) found that members of a child advocacy interdisciplinary team expressed lower levels of secondary trauma and burnout and had higher levels of compassion satisfaction. Craig and Sprang [43] (2010), who assessed a sample of 532 clinical psychologists and social workers, also found a negative association between burnout and secondary trauma to compassion satisfaction and a positive relationship between burnout and secondary trauma.

It has been suggested that resilience at work and compassion satisfaction are personal resources (the perception of the individuals that they are able to manage complex situations successfully) and, therefore, contribute toward lower levels of burnout and secondary trauma [44,45]. Two models have been proposed to explain the relationships between these four variables: the Job Demands-Resources Model (JD-R), which posits that work resources (such as participatory engagement in decision-making) contribute toward resilience, while work demands (such as high workloads) may lead to burnout [46]; and the Conservation of Resources (COR) model, which posits that individuals continuously strive to maintain their resources, but when the resources are nearing depletion, they lead to stress, secondary trauma, and burnout [47]. Resilience may be perceived as a personal resource or as a mediator for compassion fatigue and burnout [48]. 

Age and gender have both been found to impact workplace resilience, as well as levels of burnout [49,50]. Numerous studies have shown that males, as well as older personnel, report higher levels of workplace resilience compared to women and younger staff members [51,52].

The aim of the study was to assess the levels of resilience at work, burnout, secondary trauma, and compassion satisfaction among social workers in Israel during the COVID-19 pandemic and to identify associations between these variables. The hypotheses of the study were as follows: 1. Social workers will present low levels of resilience at work and compassion satisfaction during COVID-19; 2. Women will display lower levels of resilience at work and higher levels of burnout compared to men. 3. Resilience at work will be positively associated with compassion satisfaction and negatively associated with burnout and secondary trauma.

## 2. Materials and Methods

### 2.1. Study Design

The first COVID-19-confirmed patients arrived in Israel in February 2020 from the infected ship, the Princess Diamond, and by March 2020, cases of community infectivity were identified. Since then, four waves of the pandemic occurred, resulting in the morbidity of 1,354,001 confirmed cases and 8230 deaths (MoH, 2021, updated 17 December 2021). 

The study was conducted quantitatively among social workers in Israel in the midst of the COVID-19 pandemic from May to June 2020, which was during the first wave of the COVID-19 pandemic. The respondents constitute a convenience sample recruited by a snowball approach through social media by inviting social workers to respond to the online questionnaire.

The study was approved by the ethics committee of Tel Aviv University (#0001431-1 from 6 May 2020). All the participants signed an electronic informed consent form before joining the research and only then completed the questionnaire.

### 2.2. Study Population

A convenience sample of 332 social workers participated in the study. In order to be included in the study, participants had to be certified social workers with at least one year of experience. They were recruited through social media (by posts published on Facebook and in social workers’ WhatsApp groups).

### 2.3. Study Tool

The study tool was a structured questionnaire distributed online through Qualtrics software. The questionnaire included three parts: (1) Bio-demographic characteristics that included 9 items [age, gender, family, occupational status, type of fieldwork, income, contentment (self-report scale, 1–5), supervision (self-report concerning the social worker’s access to meetings with professional supervisors; dichotomic answer, yes\no), and management position (self-report dichotomic answer, yes\no)]; (2) Professional Quality of Life (ProQOL) scale in the employment realm [36]. This validated scale contains 30 items that incorporate two aspects: a positive scale (compassion satisfaction) and a negative scale called compassion fatigue (the negative scale has two categories—burnout and secondary trauma). Compassion fatigue includes feelings such as exhaustion, frustration, anger, and depression, which are typical of burnout. An example of the burnout category is: “I am not as productive at work because I am losing sleep over traumatic experience of people I help;” for the secondary trauma category: “I am preoccupied with more than one person [I help];” and an example for the compassion satisfaction category is: “I get satisfaction from being able to help people.” The ProQOL scale was used in Israel in a study with educational counselors [53]. The internal reliability of the general score was α = 0.78, burnout α = 0.79, secondary trauma α = 0.82, and compassion satisfaction α = 0.86. In another study conducted among palliative personnel, the internal reliability values were burnout α = 0.75, secondary trauma α = 0.81, and compassion satisfaction α = 0.88 [54].; and, finally, (3) Resilience at work (RAW)—a tool developed by Winwood et al. [55] (2013) to measure resilience at work for use in individual work-related performance and emotional distress contexts. The questionnaire has 7 sections: living authentically (3 items), finding your calling (4 items), maintaining perspective (3 items), managing stress (4 items), interaction cooperation (2 items), staying healthy (2 items), and building networks (2 items). The authors found that the questionnaire fits the study data, as evidenced by the fit statistics (i.e., goodness-of-fit index = 0.968; Tucker–Lewis index = 0.975; root mean square error of approximation = 0.038).

### 2.4. Data Analysis

SPSS software (version 27, Chicago, IL, USA) was used to analyze the data. As all dependent variables did not have a normal distribution, we used the Spearman correlation coefficient to examine correlations between the dependent variables—resilience at work, burnout, secondary trauma, and compassion satisfaction. In order to identify the predictors of each of the dependent variables, we first conducted a log transformation using SPSS with conversion. The prediction of each dependent variable was investigated by entering the equation variables that were found to significantly correlate with it, which included the other dependent variables (those found significantly correlated with the specifically tested dependent variable) and the bio-demographic and professional characteristics (contentment, supervision, and managing role). Scores of 61 to 80 were considered average levels of the overall resilience at work [55]. Scores of 22 or less, between 23 to 41, and 42 or more for each of the three sub-scales of the ProQOL tool, were considered as low, average, and high scores, respectively [53]. *p*-values lower than 0.05 were considered to be statistically significant. We reported the level of the correlation coefficient based on the ranking proposed by Cohen [56] )1988(. 

## 3. Results

The respondents in the study ranged from 26 to 70, with a mean age of 41.92 (SD = 10.97). The sample included 291 women and 40 men. The mean years of experience was 13.82 (SD = 10.3), ranging from 1 to 47 years of experience. The mean score of contentment was 3.30 (SD = 0.954), ranging between 1 to 5. Additional characteristics of the study sample are presented in Table 1.

### 3.1. Levels of Resilience at Work, Burnout, Compassion Satisfaction, and Secondary Trauma

The mean levels of resilience at work, burnout, compassion satisfaction, and secondary trauma were calculated. The first hypothesis was partially rejected. The overall mean level of the RAW among the social workers was found to be medium (M = 71, SD ± 8.9), as compared to the standardized scores with normative values [55]. When analyzing the varied components of the RAW, a diverse picture emerges. The mean scores for living authentically, managing stress, interacting cooperatively, and building networks were found to be higher than the normative values (+0.5–1, SD above mean). In contrast, the mean score for finding your calling was similar to the normative values of this scale (−0.5–1, SD of mean), while the mean scores of maintaining perspective and staying healthy were below the average of the normative score (−1–0.5, SD below mean). The subscales of maintaining perspective and staying healthy were in line with the hypothesis.

The average compassion satisfaction score was very close to the 50th percentile (M = 48.25); the mean scores for burnout (M = 30.18) and secondary trauma (M = 26.27) were below the 25th percentile (43 and 42, respectively). The full results are presented in Table 2.

The second hypothesis was partially verified. No significant differences were found in the average scores between genders concerning RAW, burnout, and compassion satisfaction, except for two sub-components of the RAW scale, as follows: the mean scores for females were significantly lower compared to those for males concerning finding your calling (M = 82.64, SD = 11.7 versus M = 86.87, SD = 8.3, respectively; Z = −2.235 and *p* = 0.025) and maintaining perspective (M = 35.28, SD = 14.28 versus M = 42.22, SD = 18.9, respectively; Z = −2.465 and *p* = 0.014). An opposite trend was found concerning the secondary trauma score, whereas the females’ scores were higher than those of the males’ (M = 26.76, SD = 7.54 versus M = 22.7, SD = 6.49, respectively; Z = −3.586 and *p* = 0.001).

### 3.2. Correlations between RAW, Burnout, Compassion Satisfaction, and Secondary Trauma

The third hypothesis was verified. As expected, significant correlations were found among the varied components of resilience at work, except for staying healthy, which was significantly associated only with managing stress. More interesting to note are the significantly low to medium correlations between compassion satisfaction and all the other variables, except for staying healthy. A negative association was identified between compassion satisfaction and burnout, as well as between compassion satisfaction and secondary trauma. All the other associations between compassion satisfaction and the overall RAW score, as well as with its components (as stated, except for staying healthy), were positive. Burnout demonstrated significantly low to medium negative correlations for three (out of seven) components of the RAW scale, as well as with the overall RAW score. A significant positive association was found between burnout and secondary trauma. Secondary trauma demonstrated significantly low to medium negative correlations for all components of the RAW. Table 3 presents the correlations between the study variables.

### 3.3. Prediction of Resilience at Work, Burnout, Compassion Satisfaction, and Secondary Trauma

A regression analysis was conducted to identify the factors that can predict the four study variables, including both demographic characteristics in each model as well as the varied dependent variables. The analysis revealed that high levels of compassion satisfaction, low levels of secondary trauma, high levels of contentment, and having a managerial position predicted 40.5% of the overall score of resilience at work (R^2^ = 0.405, with *p* < 0.001). Lower levels of maintaining perspective, secondary trauma, and being younger predicted 27% of burnout (R^2^ = 0.270, with *p* < 0.001). Higher levels of interacting cooperatively, finding your calling, living authentically, maintaining perspective, being older, and not being a manager predicted 58% of compassion satisfaction (R^2^ = 0.583, with *p* < 0.001). Being younger is negatively associated with secondary trauma. Higher levels of burnout, lower levels of being able to manage stress, and maintaining perspective predicted 35.9% of secondary trauma (R^2^ = 0.359, with *p* < 0.001). See Table 4, Table 5, Table 6 and Table 7.

## 4. Discussion

The COVID-19 pandemic presented a great challenge to many professionals who needed to maintain direct contact with other individuals, amongst them social workers [2]. Similar to other frontline responders, social workers were required to provide services to vulnerable populations that were severely impacted by both the risk of contracting the virus as well as the protective measures that were adopted, such as social distancing, lockdowns, and more, which enhanced their perceived isolation and loneliness [6]. The expansion of the target populations that they had to serve, in conjunction with the personal and professional challenges caused by the pandemic, led to increased stress [11,14].

In contrast to the first hypothesis, which posited that the resilience at work among social workers during the COVID-19 pandemic would be low, the study findings displayed a medium level of resilience. Social work is a helping profession, providing the social workers, as well as other professionals such as medical personnel, psychologists, or other caretakers, with a sense of accomplishment and compassion satisfaction [57,58]. This enhances the RAW, solidarity among responders, and a heightened sense of involvement in an important and fulfilling position, which are all important work resources [47,48]. These positive perceptions are similar to those of other helping professions, such as the varied medical professions, which are also characterized by increased levels of stressful events [59]. This may explain the finding that, despite the stressors of the COVID-19 pandemic, the overall mean level of resilience at work among the social workers was found to be medium. The seemingly unchanged level of resilience at work may reflect the ongoing stressors that social workers continuously manage during their work (in both routine and during adversities), which paradoxically act as protective means [57,60]. Furthermore, the national and global collaboration to contain the COVID-19 pandemic may have served as psychological capital, raising hope, optimism, and resilience, which contributed to protecting social workers from the stressors of the pandemic [31].

The second hypothesis that posited a lower level of resilience at work during COVID-19 among women was verified. This tendency may be derived from contextual or cultural perceptions of the varied roles of females versus males or result from the overall tendency of men to report higher levels of resilience compared to women. The gender differences that were found concerning finding your calling and maintaining perspective, in which females had lower levels of resilience compared to males, have also been identified in other societies and professions. Conversely, women tended to have higher levels of secondary trauma compared to males. This tendency has also been observed in the other studies [61,62,63].

In line with the third hypothesis, resilience at work was found, during the COVID-19 pandemic, to be positively associated with compassion satisfaction and negatively associated with burnout and secondary trauma. These findings are in line with both the JD-R and the COR theories, which have delineated the association between personal as well as work resources and work demands [46,47]. The depletion of resources leads to a decrease in resilience and, subsequently, to an increase in negative reactions, such as secondary trauma, stress, and burnout [48].

The analysis revealed that high levels of compassion satisfaction, low levels of secondary trauma, high levels of contentment, and having a managerial position predicted 40% of the overall score of resilience at work. Contentment, compassion satisfaction, and perceived well-being were presented in the previous studies concerning other psychotherapeutic professions as associated with resilience [64,65].

As reported previously, concerning other adversities [38], a negative association was identified during the pandemic between compassion satisfaction and burnout, as well as between compassion satisfaction and secondary trauma. All the other associations between compassion satisfaction and the overall RAW score, as well as with its components (except for staying healthy), were positive. This is in line with the previous findings that illustrated that practitioners who presented positive perceptions of their profession presented higher levels of compassion satisfaction and lower levels of secondary trauma [63,64].

Not surprisingly, burnout demonstrated significantly low to medium negative correlations with the overall RAW score. Similarly, secondary trauma demonstrated a significantly low to medium negative correlation with all the components of the RAW, except for interacting cooperatively. Lower levels of resilience at work and higher levels of secondary trauma and burnout have been shown to decrease the quality of care that social workers provide to their clients and weaken the therapeutic results [28,60,64,66]. The self-care of social workers is, thus, crucial in order to prevent burnout and secondary trauma and to maintain resilience at work [57,63]. As demonstrated in the other studies, a significant positive association was found during the COVID-19 pandemic between burnout and secondary trauma [40,41]. As previously published, it is vital to assess the subjective well-being and levels of burnout of therapists, as these impact not only their work resilience but also the results of their therapeutic work [28].

In contrast to the previous findings, seniority (number of years in the workplace) in social work during the COVID-19 pandemic was not found to predict RAW and burnout [67,68]. This may be derived from the innovative nature of the COVID-19 pandemic. It is a novel experience for all social workers, regardless of their years of experience. Furthermore, we conjecture that, though senior social workers were found to have lower levels of burnout in routine compared to more junior ones, as COVID-19 posed a higher health risk to the older ones, this may have impacted their concerns for their own well-being and, thus, contributed to a higher perceived threat and burnout symptoms.

There are two main theoretical implications of this study: The first is the centrality of the variable RAW and its impact on burnout, secondary trauma, and compassion satisfaction. To the best of our knowledge, the research concerning this issue is scarce. The second is that ongoing work demands, most especially loaded and stressful work conditions, may actually serve, during severe adversities, as a protective measure against a drastic decrease in resilience at work and compassion satisfaction. This conjecture should be further evaluated to understand whether it can be identified concerning other professions and/or other types of adversities.

## 5. Limitations and Future Research

Three main limitations should be noted regarding the study. The study was conducted as cross-sectional research and, thus, it is not possible to make any causal inferences. The use of convenience sampling does not enable us to make generalizations of the results of the study to the overall social work population due to the possibility of bias and the under- or over-representation of specific groups of social workers. The third limitation is that data were collected at a one-time frame, which may reflect specific (and temporary) feelings and perceptions that each respondent felt, regardless of COVID-19. This might affect the exactness of the research conclusions.

Further studies are warranted to build an evidence-based body of knowledge that will focus on the centrality of RAW and its impact on burnout, secondary trauma, and compassion satisfaction. We recommend that prospective longitudinal studies be conducted, which will facilitate follow-up over time, cluster sampling, and in-depth monitoring of the association between resilience at work and burnout among frontline workers.

## 6. Conclusions

Social workers are vital frontline workers in both routine times and during adversities; they cater to the needs of the most vulnerable sects of the population. Considering their importance, it is imperative that social workers be part of the overall response to varied types of emergencies. Nonetheless, ensuring the effective participation of social workers in the overall response to any type of known or emerging threat necessitates empowering them to enhance their resilience at work and mitigate their levels of burnout. 

As the COVID-19 pandemic still challenges most societies and, furthermore, additional adversities (both human-made or those resulting from climate change or other natural events) are expected, policymakers should consider ways to integrate ongoing mechanisms that will enhance the resilience at work of all therapeutic professionals, including those of social workers. These measures include better work conditions, higher availability of resources, and extended budgets [9,10]. Considering the positive association found between resilience at work and compassion satisfaction, investing efforts in the well-being of social workers is expected to substantially enhance their capacity to contribute toward a holistic emergency response to varied crises.

In order to generalize the findings of this study to varied frontline personnel, longitudinal studies should be conducted to identify commonalities and diversities concerning the well-being, workplace resilience, and burnout among different healthcare and welfare workers.

## Figures and Tables

**Table 1 ijerph-19-05500-t001:** Characteristics of the study population (*n* = 332).

Characteristic		Percentage (*n*)
Status of employment	Salaried employee	93.1% (309)
	both salaried and self employed	2.4% (8)
Supervision (ongoing)	Yes	98.7% (228)
	No	
Management status	Yes	39.5% (131)
	No	
Marital status	Married	77.4 (257)
	Single	14.2 (47)
	Divorced	6.9 (23)
	Widowed	1.5 (5)
Personal income	below average	33.1 (110)
	close to average	53.9 (179)
	above average	6 (20)
Household income	below average	33.1 (110)
	close to average	45.2 (150)
	above average	10.5 (35)

**Table 2 ijerph-19-05500-t002:** Average scores of resilience at work and ProQOL scale; (*n* = 332).*

Variable Name	Mean	Standard Deviation (SD)	Minimum Score	Maximum Score
**Resilience at Work**				
Living authentically	86.31	8.35	38.89	100
Finding your calling	83.19	11.42	33.33	100
Maintaining perspective	36.12	15.03	5.56	72.22
Managing stress	69.32	16.78	16.67	100
Interacting cooperatively	77.96	13.3	25	100
Staying healthy	61.49	22.07	16.67	100
Building networks	79.92	17.7	16.67	100
Overall RAW	70.74	8.39	40.83	89.17
**Professional Quality of Life scale**				
Burnout	30.18	5.93	16	51
Compassion satisfaction	48.25	7.06	22	60
**Secondary trauma**	**26.27**	**7.52**	**12**	**51**

* Standard scores of 61 to 80 are considered as ‘average’ RAW scores [55]; standard scores of 23 to 41 are considered as ‘average’ ProQOL scores in each of the three sub-scales [53].

**Table 3 ijerph-19-05500-t003:** Spearman coefficient correlation between all dependent variables (*n* = 332).

	Compassion Satisfaction	Burnout	Secondary Trauma	Living Authentically	Finding Your Calling	Maintaining Perspective	Managing Stress	Interacting Cooperatively	Staying Healthy	Building Networks
Compassion satisfaction										
Burnout	−0.15 **									
Secondary trauma	−0.32 **	0.37 **								
Living authentically	0.54 **	0.08	−0.20 **							
Finding your calling	0.64 **	−0.16 **	−0.28 **	0.44 **						
Maintaining perspective	0.37 **	−0.38 **	−0.50 **	0.20 **	0.29 **					
Managing stress	0.32 **	−0.24 **	−0.33 **	0.28 **	0.19 **	0.41 **				
Interacting cooperatively	0.26 **	−0.04	−0.07	0.19 **	0.17 **	0.11 *	0.20 **			
Staying healthy	0.08	−0.04	−0.13 *	0.03	0.02	0.08	0.23 **	−0.07		
Building networks	0.33 **	−0.05	−0.13 *	0.17 **	0.42 **	0.16 **	0.22 **	0.20 **	0.16 **	
RAW	0.57 **	−0.28 **	−0.45 **	0.46 **	0.56 **	0.63 **	0.75 **	0.35 **	0. 41 **	0.53 **

* *p* ≤ 0.05, ** *p* ≤ 0.01

**Table 4 ijerph-19-05500-t004:** Regression analysis; dependent variable: RAW, R ^2^ = 0.405, *p* < 0.001.

Variable	B	SE B	β	Sig
Age	0.000	0.000	−0.066	NS
Gender	0.007	0.007	0.045	NS
Contentment	0.009	0.003	0.152	0.002
Supervision	0.002	0.004	0.020	NS
Management position	0.012	0.005	0.107	0.020
Income	−0.001	0.003	−0.001	NS
Professional status	−0.006	0.006	−0.045	NS
Compassion Satisfaction	0.356	0.038	0.469	0.001
Burnout	−0.040	0.030	−0.065	NS
Secondary trauma	−0.099	0.022	−0.225	0.001

B = unstandardized B, SE B = coefficients Std Error, β = standardized coefficients Beta, Sig = Significant.

**Table 5 ijerph-19-05500-t005:** Regression analysis; dependent variable: burnout, R^2^ = 0.270, *p* < 0.001.

Variable	B	SE B	β	Sig
Age	−0.001	0.000	−0.127	0.021
Contentment	−0.008	0.005	−0.087	NS
Supervision	0.012	0.007	0.077	NS
Management position	−0.004	0.009	−0.020	NS
Income	0.005	0.006	−0.048	NS
Compassion Satisfaction	0.026	0.081	0.021	NS
Secondary trauma	0.150	0.041	0.211	0.001
Finding your calling	−0.054	0.080	−0.402	NS
maintaining perspective	−0.078	0.024	−0.194	0.002
Managing stress	−0.071	0.037	−0.103	NS

Sig = Significant.

**Table 6 ijerph-19-05500-t006:** Regression analysis; dependent variable: compassion satisfaction, R^2^ = 0.583, *p* < 0.001.

Variable	B	SE B	β	Sig
Age	0.001	0.000	0.094	0.024
Contentment	0.004	0.003	0.049	NS
Management position	−0.013	0.006	−0.092	0.018
Income	0.002	0.004	0.022	NS
Burnout	−0.035	0.035	−0.042	NS
Secondary trauma	−0.018	0.026	−0.031	NS
Living authentically	0.487	0.067	0.316	0.001
Finding your calling	0.381	0.047	0.368	0.001
maintaining perspective	0.038	0.015	0.117	0.012
Managing stress	−0.009	0.024	−0.016	NS
Interacting cooperatively	0.118	0.035	0.139	0.001
Building networks	0.005	0.023	0.009	NS

Sig = Significant.

**Table 7 ijerph-19-05500-t007:** Regression analysis; dependent variable: secondary trauma, R^2^ = 0.359 *p* < 0.001.

Variable	B	SE B	β	Sig
Age	−0.002	0.001	−0.192	0.041
Contentment	−0.003	0.007	−0.022	NS
Years of experience	0.017	0.010	0.001	NS
Supervision	0.017	0.010	0.001	NS
Management position	0.011	0.012	0.043	NS
Compassion satisfaction	−0.094	0.123	−0.055	NS
Burnout	0.233	0.077	0.167	0.003
Living authentically	−0.060	0.161	−0.023	NS
Finding your calling	−0.126	0.112	−0.072	NS
maintaining perspective	0.183	0.032	−0.328	0.001
Managing stress	−0.142	0.053	−0.145	0.008
Interacting cooperatively	0.058	0.080	0.040	NS
Staying healthy	−0.033	0.031	−0.053	NS
Building networks	0.096	0.050	0.102	NS

Sig = Significant.

## Data Availability

The data accumulated in the study are kept by the authors. Data are not published openly due to privacy issues, but the analyzed data are available from the authors upon request.

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
