# Peer review of "Resilience at Work, Burnout, Secondary Trauma, and Compassion Satisfaction of Social Workers Amidst the COVID-19 Pandemic"

_ijerph, 2022, doi:10.3390/ijerph19095500_

Round 1
Reviewer 1 Report
- In the abstract, please specified how many female participants were tested in this study.
- The article is missing important information for the reader, like, the theoretical framework of the relationship between variables. The authors presented only the empirical results as arguments for relationships between variables. It is necessary to develop the theoretical argumentation of this study
- What are your hypothesis or research questions?
- The authors did not provide detailed information regarding participant recruitment, which is an important part of a scientific paper.
- Instruments: please provide proof about the adaptation and validation of this scale on the Israelite population or made CFA on each scale. All the data were collected from respondents at one time, which may affect the accuracy of the research conclusions. Hence, it would help if you did CMC before testing the hypothesizes. Also, please provide examples of items to each scale and information about internal consistency.
- Where is the table with the results of the regression analysis?
- In the Discussion section, please follow the logic of each hypothesis and discuss the results of each hypothesis in relation to theoretical perspective and empirical studies.
- What are the theoretical implications of your study?
- It is needed to create a new section Limits and future research and develop part of future research.
Reviewer 2 Report
This is a review of the article, “Resilience at Work, Burnout, Secondary Trauma and Compassion Satisfaction of Social Workers amidst the COVID-19 Pandemic” for the International Journal of Educational Research and Public Health.
The article certainly has merit and could add to the literature, as it examines the relationship of the three Professional Quality of Life Sub-Scales (by Stamm) with each other, with the Resilience at Work Scale (and sub-scales), and demographic/professional factors for a sample of 332 social workers in Israel during the first surge of the COVID-19 pandemic in May-June 2020.
However, there are many aspects of the article that seem rushed, both in the presentation of data and in the writing. My numerous comments and criticisms are as follows:
For the Abstract, the authors should note when the study took place. They should indicate that data collection occurred from May to June 2020 during the first wave of the COVID pandemic. In paragraph 2, the Professional Quality of Life Scale should be cited by its proper name, not “Quality of Life at work”; and the word “scale” should be added to both measures cited. The phrase “during the COVID-19” needs to be corrected. Capitalizations should be used consistently. Sometimes the same concepts are capitalized, sometimes not. Similarly, sometimes the authors use double quotation marks, and sometimes single ones. There needs to be consistency here.
In regard to the Abstract, there is no reporting of the results of the bivariate analysis even though the Results section of the article seems to provide more details of these findings (including a Table) than for the multivariate analysis. In the Abstract, there is only reporting of the results of the multivariate analysis, which seems to garner less attention in the article with no Table. In fact, the Abstract does not indicate that both bivariate and multivariate analyses were performed.
Also in the Abstract, the authors do not indicate that the two major scales that they utilized have sub-scales. However, they report the results of the analyses of the sub-scales. This is very confusing.
Grammar in the article needs more attention. Two examples of errors in the Introduction are “provide an optimal care,” “codes of ethic.”
In the Introduction, on line 93 of page 2, the authors write that Ben-Porat “found an average level of secondary traumatization among workers.” This does not really mean anything.
Is this an average score on the Stamm sub-scale of “secondary traumatic stress” compared to
other healthcare professionals? What does this mean? This is the same issue as on page 3, line 102 in regard to “secondary trauma at an average level” and page 3, line 125, regarding “average levels of burnout.” What does an average level of burnout mean? This needs to be explained.
The authors need to use the term “psychotherapy” on page 2, line 123, not just “therapy.”
On page 4, there are some grammar errors in “Study Design,” for example: “The study was a quantitative research…” and “at May to June 2020.” Also, please use double quotation marks, not single ones.
In the Study Design section, it states that the study used a “snowball approach.” However, it seems that most of the design was “convenience” or “availability” sampling, as is stated on line 168 of page 4, although there may have been some snowballing.
The 2.3 “Study Tool” section (page 4) needs some more clarity. What do the authors mean by “based on the ProQOL Professional Quality of Life) scale?” Did they use the scale, or did they change it in some way? They list the three sub-scales that they used rather vaguely, naming one “Compassion Fatigue.” In general, the three sub-scales of the ProQOL scale are referred to as “Compassion Satisfaction,” “Burnout,” and “Secondary Traumatic Stress.” In the rest of the article, the authors use the term, “Burnout.” So why are they calling it “Compassion Fatigue” here. The authors should list the titles of these three important sub-scales clearly here. They form the basis of much of the article. And they should be sure to list these with the same name throughout the article.
In 2.4 (Data Analysis) (page 4), the authors discuss how they examined the correlation between the dependent variables: resilience at work, burnout, and compassion satisfaction. What happened to “secondary traumatic stress?”
In 2.4, (top of page 5), they refer to professional characteristics they measured: “contentment, supervision, and managing-role.” How were these measured? What exactly was measured? For example, in the Results section on page 5, it refers to the “mean score of contentment.”
The authors need to clarify what this means.
The bivariate results of the study are presented in 3.1 on page 5. Some of the variables are capitalized, some not. Some have quotation marks around them, some not. The authors need to be consistent here.
The Results section on page 5 begins with describing the characteristics of the sample. After a number of these statistics, the article then states that additional statistics are shown in Table 1.
However, two out of the six statistics in Table 1 are already described in the initial paragraph.
I think it would make more sense to list all of the characteristics in Table 1.
Table 2 on page 6 shows the mean values of the different sub-scales. The table is confusing in that the seven sub-scales of the Resilience at Work (RAW) scale are presented first, with the overall RAW score below it. Then the results of the three Professional Quality of Life sub-scales are listed, without any reference to that composite title. Some of the words in the names of the scales are capitalized, some not.
It would be important to add a column to list the categories of mean scores of each of the sub-scales and scales in Table 2 from other studies of those scales, as the authors refer to these
“averages” throughout the article.
The title of 3.2 of the Results section on page 6 states “secondary relocation.” Is that supposed to be “secondary traumatization?”
It would be good if the authors could be consistent throughout the article in naming the variables. For example, the variable “secondary traumatic stress” is also referred to as “secondary trauma” and “secondary traumatization.”
In Section 3.2 of the Results section on page 6, the authors state “Secondary trauma demonstrated significant low to medium negative correlation for all components of the RAW…”
This terminology regarding level of “correlation” is then used frequently throughout the article.
But the authors do not state what this means. Are these effect sizes? Please be clear. There is no discussion of effect sizes in the Analysis section.
The bivariate results are shown on Table 3 on page 7. Once again, some variables have words capitalized, and some do not. The Spearman Coefficient Correlations are all listed. Those that demonstrate statistically significant differences have either one or two asterisks after them, but the Table does not state what this means. In the article, it states that the p value for significance is 0.05. What do the asterisks mean?
Section 3.3 on page 8 delineates the findings of the multivariate analysis of the study. It is short and rushed with no Table. Results seem to be chosen at random. The first sentence discusses “the four study variables,” but many sub-scales of the RAW are analyzed. There are typos/grammar errors here. For example, it states “less containment” on line 261, which I assume is “less contentment.” This section really needs to be organized in a structured way with a Table added. It is not acceptable as it is.
In the Discussion section on page 8 on lines 282, the authors discuss “solidarity among responders” and “a heightened sense of involvement in an important and fulfilling position.” I have no idea where these concepts came from.
The Discussion section is weak in relating the article to the COVID-19 pandemic. There is no real discussion how this study took place during the very first wave of the pandemic when
so much about COVID was unknown. When reading the Discussion section, the COVID situation almost seems tangential to the study’s examination of the relationship between the variables.
There is a typo on Line 298 before “40%.”
On page 12 of the Discussion, the authors discuss how “in contrast to previous findings, seniority in social work was not found to predict the RaW [incorrect small “a”] and burnout.”
First of all, I am not sure what variable(s), “seniority” is referring to. Secondly, this part of the Discussion fails to note that from the reporting of the multivariate analysis results on page 8 in 3.3, “being younger predicted 36% of the secondary traumatization” and also predicted 27% of burnout. There were also statistically significant results regarding being and not being a manager in 3.3. Do these pertain to seniority as well?
In conclusion, this study contains many interesting results. However, the article needs considerable work regarding its organization, presentation of data, and in quality of writing.
If the authors carefully revise the article, it should be publishable.
Round 2
Reviewer 1 Report
For updated information related to new research, please consult these references
Vîrgă, D., Baciu, E. L., Lazăr, T. A., & LupÈ™a, D. (2020). Psychological capital protects social workers from burnout and secondary traumatic stress. Sustainability, 12(6), 2246.
In the Results section, it is needed to present the results related to each hypothesis and explain if the hypothesis is confirmed or not by the results.
Also, in Limits section, it needs to propose a future solution for each limit identified in this paper.
Reviewer 2 Report
This is a review of the revised version of the article, “Resilience at Work, Burnout, Secondary Trauma and Compassion Satisfaction of Social Workers amidst the COVID-19 Pandemic” for the International Journal of Educational Research and Public Health.
There has been considerable improvement in this version of the article from the first submission. The article certainly has worthwhile information and could be publishable.
However, there are still some problems within the article that need to be addressed:
- Overall, there are still many grammar issues and typos within the article. At the end of this review, I will provide a cursory list of these errors, but there are more of them. The authors need to proofread the article before resubmitting it.
- Abstract:
- a) In the second paragraph (line 23), it does not state that “maintaining perspective” and “staying healthy” are part of the RAW.
- b) In the second paragraph (line 23-25), it does not state that “compassion satisfaction,”
“burnout” and “secondary trauma” are sub-scales of the Professional Quality of Life scale.
The reader would not know what scale these three sub-scales belong to.
- Page 3, Line 102: There is no mention of what “secondary trauma stress scale” the authors are referring to. Please clarify this. The sentence also has multiple grammar errors.
- Page 4, Lines 168-173. This version of the article now lists three hypotheses. They are hastily introduced. The second one “Women will display lower levels of resilience at work and higher levels of burnout compared to men” comes out of nowhere. There is no discussion of such an issue in the literature review. Why is this taking such a prominent position? The authors should add content to the literature review to highlight the importance of this hypothesis, or eliminate it as a hypothesis. There are numerous findings in the study that are not reflected in the hypotheses. It is not clear why this one is given importance here without any prior discussion about the issue.
- Page 4, Line 197. “Supervision” is listed as a variable (yes/no). But it is not clear what this means. Are the researchers asking whether or not the participant had supervision at all?
Please clarify.
- Page 5, line 207. The word “scale” appears in the middle of the sentence for no reason.
- IMPORTANT: In the first version of the article, I noted that Table 2 on Page 6 should include the standard levels (low, medium, high) for each of the variables so that the reader could compare the results of this study with the standard levels. They have not done so. Therefore, the scores presented are meaningless to the reader. The authors need to include this.
- In the Discussion section on Page 11, Lines 320-329, the authors note that their first hypothesis was not confirmed, and then spend nine lines explaining the reasons for their hypothesis. I think this is confusing and unnecessary. These lines should be eliminated, as the next paragraph conjectures why the hypothesis was not validated. This is all that is necessary here. The literature review should explain the author’s original hypothesis. There is no need to explain it here, as it was not reflected in the results.
- I do not understand the sentence on Page 11, lines 350 and 351: “Nonetheless, gender differences were not the main element that was found as associated with resilience at work.”
This seems to contradict what is written on page 11, Lines 342-343. What do the authors mean here?
- For Page 12, Lines 386-387, are the authors referring to previous studies when they write:
“Furthermore, though senior social workers were found to have lower levels of burnout in routine compared to more junior ones…” If so, please offer citations here. If the authors mean something else, please clarify.
- For page 12, Lines 403-404, please expand on the meaning of the third limitation: “The third limitation is that data were collected at one time, which may affect the accuracy of the research conclusions.” How is this a limitation?
- Page 12, Lines 405-407, there is a sentence that stands out by itself for no reason. Please fix this. (It starts with “Further studies are warranted…)
- The section “Conclusions” seems very rushed and superficial. After all of the detailed information presented in this article, can it have a more meaningful Conclusion?
- On Table 4.4, why is .057 for “Building networks” considered statistically significant?
HERE IS A LIST OF SOME OF THE GRAMMAR/TYPO ERRORS. HOWEVER, THIS LIST IS NOT EXHAUSTIVE, AND THE AUTHORS NEED TO CAREFULLY PROOFREAD THE ARTICLE.
page 2, line 43: no “a” needed
page 2, line 63: no “the” needed
page 2, line 65: no comma before “required”
page 2, line 91: “at risk”; not “in risk”
page 2, line 93: “a higher risk of mental health problems” in relation to what?
page 3, line 99: no comma before “indicated”
page 3, lines 102-103—sentence fragment
page 3, line 105, Add “the” before “ProqOL scale.” Eliminate comma after this.
page 3, line 114: “it being” not “its being”
page 3, line 117: Eliminate “the” before “patients.”
page 3, line 128: Eliminate “the” before “COVID-19.”
page 3, line 139: “Senreich et al. (2020) that assessed 6,112 social workers”---has poor grammar
Please rephrase.
page 3, line 146: Add comma after “secondary trauma” and eliminate comma after “contrast.”
page 4, line 149: Eliminate “that” before the word “expressed.”
page 5 line 207: Remove the word “scale” floating in the middle of the line.
page 5, lines 213-219: Why is the name of one sub-scale capitalized, but the rest not?
Please be consistent. This is true for other lists throughout the article.
See page 6, line 241 for example.
page 5, line 235---Add a parenthesis after 10.97.
page 5, line 260: “me” should be “m”
page 9, line 282: “Resilience at Work” not “Resilience of Work.”
page 9, line 289: Remove “the” before “burnout.”
page 9, line 291: Remove “the” before “compassion satisfaction.”
page 9, line 291-292: “Being younger is more likely to have secondary trauma” is grammatically incorrect.
page 9, line 293: “higher levels of building networks” instead of “higher ability of building networks”
page 11, line 319: “would be low” rather than “will be low”
page 11, line 325: “frontline responders” instead of “frontal responders.”
page 12, line 373: There are two periods after “cooperatively.” Please eliminate one.
page 12, line 400: Remove “a” before “convenience sampling.:
page 12, line 400-401: The grammar of the sentence is incorrect “…does not enable to generalize the results of the study…”
